In-silico study of antisense oligonucleotide antibiotics

Chen Erica S.
http://orcid.org/0000-0002-0713-9545 Ho Eric S. hoe@lafayette.edu
Biology, Lafayette College , Easton, PA , United States
Dong Peixin
Electronic publication date: 2023 Nov 15
Publication date: 2023
Volume: 11
Electronic Location ID: e16343
Received 2023 Jul 4; Accepted 2023 Oct 3
Copyright: © 2023 Chen and Ho
Copyright year: 2023
Copyright holder: Chen and Ho
License: This is an open access article distributed under the terms of the Creative Commons Attribution License, which permits unrestricted use, distribution, reproduction and adaptation in any medium and for any purpose provided that it is properly attributed. For attribution, the original author(s), title, publication source (PeerJ) and either DOI or URL of the article must be cited.
License URL: https://creativecommons.org/licenses/by/4.0/

Keywords: Bioinformatics, Antisense oligonucleotide, Antibiotic, Antibiotic resistance

Funding: Lafayette College This work was supported by Lafayette College. The funders had no role in study design, data collection and analysis, decision to publish, or preparation of the manuscript.

==============================
Background

The rapid emergence of antibiotic-resistant bacteria directly contributes to a wave of untreatable infections. The lack of new drug development is an important driver of this crisis. Most antibiotics today are small molecules that block vital processes in bacteria. To optimize such effects, the three-dimensional structure of targeted bacterial proteins is imperative, although such a task is time-consuming and tedious, impeding the development of antibiotics. The development of RNA-based therapeutics has catalyzed a new platform of antibiotics—antisense oligonucleotides (ASOs). These molecules hybridize with their target mRNAs with high specificity, knocking down or interfering with protein translation. This study aims to develop a bioinformatics pipeline to identify potent ASO targets in essential bacterial genes.

Methods

Three bacterial species (P. gingivalis, H. influenzae, and S. aureus) were used to demonstrate the utility of the pipeline. Open reading frames of bacterial essential genes were downloaded from the Database of Essential Genes (DEG). After filtering for specificity and accessibility, ASO candidates were ranked based on their self-hybridization score, predicted melting temperature, and the position on the gene in an operon. Enrichment analysis was conducted on genes associated with putative potent ASOs.

Results

A total of 45,628 ASOs were generated from 348 unique essential genes in P. gingivalis. A total of 1,117 of them were considered putative. A total of 27,273 ASOs were generated from 191 unique essential genes in H. influenzae. A total of 847 of them were considered putative. A total of 175,606 ASOs were generated from 346 essential genes in S. aureus. A total of 7,061 of them were considered putative. Critical biological processes associated with these genes include translation, regulation of cell shape, cell division, and peptidoglycan biosynthetic process. Putative ASO targets generated for each bacterial species are publicly available here: https://github.com/EricSHo/AOA. The results demonstrate that our bioinformatics pipeline is useful in identifying unique and accessible ASO targets in bacterial species that post major public health issues.

Introduction

Antibiotics are foundational in modern medicine, playing essential roles in treating various infections. However, rapidly emerging antibiotic-resistant bacteria are increasingly common, leading to a wave of untreatable infections (Blair et al., 2015). Conditions such as pneumonia, tuberculosis, and foodborne diseases are harder (in cases impossible) to treat due to the ineffectiveness of antibiotics (Ventola, 2015). The crisis can be traced back to three main causes: overuse and misuse of antibiotics, lack of new drug development resulting from reduced economic incentives, and challenging regulatory requirements (Ventola, 2015).

Most antibiotics today are small molecules discovered between the 1940s and 1960s (Lewis, 2013). These compounds block vital processes in bacteria, ultimately hampering growth. Antibiotics generally target DNA replication, RNA synthesis, cell wall synthesis, and protein synthesis (Kohanski, Dwyer & Collins, 2010). One of the approaches to discovering drug candidates is by determining the three-dimensional structure of the target bacterial protein. To do so, chemists may determine the three-dimensional structure of the targeted bacterial protein through purification, followed by X-ray crystallography or NMR. However, such a task is time-consuming and tedious. Even with the recent successes accomplished by AI-based protein folding prediction methods, such as AlphaFold (Jumper et al., 2021), and RoseTTAFold (Baek et al., 2021), it only solves part of the question as the binding pockets and the small molecules remain to be resolved. Meanwhile, the idea of leveraging bacteriophages (phages) against bacteria dates back to the seminal work of Felix D’Herelle. However, isolating the phages specific to bacteria is a daunting task. Recently, there have been efforts to utilize antisense peptide nucleic acids (PNAs) as antibiotics (Popella et al., 2022). A web service has been published to predict PNA binding sites (Jung et al., 2023). However, the prediction of PNA binding sites is still nascent as the rules governing binding are less developed than DNA/RNA hybridization. Thus, improvement in antibiotic development is imperative to keep up with the increased presence of antimicrobial-resistant bacteria.

Importantly, many antibiotics prescribed today are considered broad-spectrum antibiotics. These drug compounds target a wide range of Gram-positive and Gram-negative bacteria (Grada & Bunick, 2021). However, such antibacterial agents are neither specific nor effective. There are two main issues with broad-spectrum antibacterial agents—antimicrobial resistance and off-target effects. The lack of specificity, especially for Gram-negative bacteria, harms beneficial bacterial species critical to good health (Drawz & Bonomo, 2010; Howard, Gonzalez & Garneau-Tsodikova, 2021).

Since the rise of antibiotic-resistant microorganisms, antimicrobial peptides (AMPs) appear to be a promising platform for antibiotics. Such compounds are small bioactive proteins that are naturally produced by all organisms and serve as the first line of defense against various pathogens (fungi, viruses, and bacteria) (Moretta et al., 2021). AMPs are applicable to agriculture, the food industry, and medicine (Moretta et al., 2021).

RNA-based therapeutics

mRNA is an intermediate product of the central dogma, bridging the gap between genetic information and biological function, although non-coding RNAs also perform functions. The concept of mRNA-based drugs began in 1989 and has since been a rapidly developing area in drug development (Xu et al., 2020). mRNA-based vaccines have demonstrated potential in inducing immunological results against cancers and various infectious diseases, the most notable being the Pfizer-BioNTech and Moderna vaccines against SARS-CoV2 (Heine, Juranek & Brossart, 2021).

Such vaccines encode immunogenic targets and are expressed by antigen-presenting cells, eliciting a strong immune response (Miao, Zhang & Huang, 2021). Furthermore, mRNA-based vaccines demonstrate several advantages over conventional vaccine platforms. These include high potency, increased specificity, flexibility, a strong induced immune response, and cost-effectiveness (Xu et al., 2020).

Antibiotics

Development in mRNA vaccines and RNA-based therapeutics catalyzed a new platform of antibiotics—RNA-based antibiotics. One class of DNA/RNA-based antibiotics is known as antisense oligonucleotides (ASOs) due to their complementary sequences. ASOs bind to microbial mRNA, preventing translation in proteins (Hegarty & Stewart, 2018). Conceptually, ASOs mitigate the two main issues of traditional antibiotics due to their specificity and ease of manipulation.

ASOs typically play a role in gene silencing and prevent translation via two paths. The first means of preventing translation is through steric blockage of ribosomal subunit assembly and access to mRNA binding sites (Hegarty & Stewart, 2018). The second is through mRNA degradation. An ASO gapmer bound to a complementary transcript is recognized by RNase H, an endoribonuclease, leading to mRNA degradation (Hegarty & Stewart, 2018). ASOs are chemically modified to increase stability and specificity. Modifications include ribose sugar, nucleic acid backbone, and nucleobase (Wan & Seth, 2016; Geary et al., 2015). Currently, the only FDA-approved ASO therapeutic that targets microorganisms is fomivirsen (brand name Vitravene). Fomivirsen was approved in 1998 for the treatment of cytomegalovirus-induced retinitis in HIV-positive patients (Li et al., 2022). However, Vitravene was withdrawn from the EU in 2002 and the US in 2006 due to the lack of patients (Bubela & Christopher McCabe, 2014). Nevertheless, ASOs provide a promising platform to combat antibiotic-resistant bacterial strains and increase drug specificity. This study aims to discover unique and accessible ASO binding sites in three different bacteria using a bioinformatics pipeline.

Bacteria of interest

Porphyromonas gingivalis

P. gingivalis is a Gram-negative anaerobic bacterium. It is a critical etiologic agent contributing to periodontitis (Prakasam, Elavarasu & Natarajan, 2012). Out of the over 500 bacterial species present in human subgingival plaque, P. gingivalis stands out as a keystone species in disease development (Prakasam, Elavarasu & Natarajan, 2012). Although present at low levels in healthy individuals, P. gingivalis has a prevalence of 78% [95% CI [71–84]] in periodontal disease groups (Rafiei et al., 2018).

P. gingivalis mainly inhabits the subgingival sulcus of the human oral cavity, colloquially referred to as the “periodontal pocket” (Prakasam, Elavarasu & Natarajan, 2012). Studies show that P. gingivalis exhibits metabolic plasticity, enabling it to use non-proteinaceous substrates for colonization and biofilm formation (Moradali & Davey, 2021). P. gingivalis expresses a variety of virulence factors, including aminopeptidases and lipopolysaccharides (Zheng et al., 2021). These factors overturn the host’s immune responses and lead to chronic inflammation in infected tissues (Zheng et al., 2021).

In practice, periodontists only use antibiotics alongside mechanical debridement in cases with severe progressive periodontitis. Because systemic antibiotics are used, their application is restricted to severe cases (Conrads et al., 2021). Commonly prescribed antibiotics include metronidazole, amoxicillin, and clindamycin (Conrads et al., 2021). However, certain fluoroquinolones (mostly ciprofloxacin), tetracyclines, and macrolides (such as azithromycin) are also prescribed (Conrads et al., 2021). See Table 1 for the biological pathways targeted by these antibiotics.

Table 1 Antibiotics and targeted biological pathways from DrugBank (https://go.drugbank.com/).

Antibiotic	Target biological pathway	DrugBank ID	
Metronidazole	Inhibition of protein synthesis via interaction with DNA	DB00916	
Amoxicillin	Inhibition of transpeptidation via binding to penicillin-binding proteins (PBPs)	DB01060	
Clindamycin	Inhibition of peptide bond formation via binding to 50S ribosomal subunit	DB01190	
Ciprofloxacin	Inhibition of DNA synthesis by targeting DNA gyrase and topoisomerase IV	DB00537	
Tetracycline	Inhibition of protein synthesis via preventing the association of aminoacyl-tRNA with bacterial ribosome	DB00759	
Azithromycin	Inhibition of protein synthesis via binding to 50S ribosomal subunit	DB00207	
Cefazolin	Disrupt cell-wall synthesis via beta-lactam rings that bind to PBPs	DB01327	
Azithromycine	Inhibition of protein synthesis by binding to the 23S rRNA of the 50S ribosomal subunit	DB00207	
Doxycycline	Inhibition of protein synthesis via allosteric binding to the 30S ribosomal subunit	DB00254	
Meropenem	Inhibition of peptide cross-linking and peptidase reactions through binding to PBPs	DB00760	
Nafcillin	Inhibit cell wall synthesis via binding to PBPs	DB00607	
Oxacillin	Inhibit cell wall synthesis via binding to specific PBPs within the cell wall and inhibiting the third and last stage of cell wall synthesis	DB00713	
Vancomycin	Inhibit cell way synthesis via preventing the incorporation of NAM and NAG peptide subunits into the peptidoglycan
matrix	DB00512	
Daptomycin	Mechanism isn’t well understood; studies point to inhibition of cell membrane and cell wall synthesis	DB00080	
Linezolid	Inhibit protein translation via binding to a site on the bacterial 23S ribosomal RNA	DB00601	
Note:

Approved antibiotics, target pathways, and IDs from Drugbank.ca.

Haemophilus influenzae

H. influenzae is a Gram-negative facultatively anaerobic bacterium (Khattak & Anjum, 2022). It contributes to various diseases; the most prominent being lower respiratory tract infections (such as pneumonia), septic arthritis, and sinusitis (Khattak & Anjum, 2022). Although H. influenzae is present in the noses and throats of healthy individuals, it can lead to infections if the bacterium travels to other parts of the body (Centers for Disease Control (CDC), 2022).

There are six distinct types of H. influenzae (a through f), however, they can be generally classified as encapsulated or non-encapsulated types (Khattak & Anjum, 2022). H. influenzae exhibits a variety of virulence factors including its capsule, adhesion proteins, pili, outer membrane proteins, IgA1 protease, and lipooligosaccharide (Kostyanev & Sechanova, 2012). Moreover, non-encapsulated strains exhibit various virulence factors that have been an area of focus in the post-vaccine area (Kostyanev & Sechanova, 2012). These factors actively participate in the host invasion process.

Antibiotics used in the treatment of H. influenzae cases include amoxicillin, cephalosporin (such as cefazolin), azithromycin, doxycycline, fluoroquinolones, and carbapenems (such as meropenem) (Vaez et al., 2019). See Table 1 for the biological pathways targeted by these antibiotics. The increasing incidence of resistance to aminopenicillins (bactericidal beta-lactam antibiotics) is a cause of concern globally. In fact, the WHO placed ampicillin-resistant H. influenzae strains at medium priority (in terms of urgency for antibiotic development) in their list of antibiotic-resistant bacteria (Vaez et al., 2019). The bacterium’s antibiotic resistance characteristics are induced by enzyme mechanisms as well as changes in targets (Kostyanev & Sechanova, 2012).

Staphylococcus aureus

S. aureus is a Gram-positive facultatively anaerobic bacterium. S. aureus was selected for the study because it is one of the leading causes of bacterial infections in humans (Somerville & Proctor, 2009). It contributes to diseases including bacteremia, endocarditis, and infections of both skin and soft tissue (Taylor & Unakal, 2022). S. aureus is found on the skin and mucous membranes of up to half of all adults (Taylor & Unakal, 2022).

S. aureus is a prevalent species in the discussion about antibiotic-resistant microorganisms. The pathogen can be classified into two kind strains: methicillin-sensitive S. aureus (MSSA) and methicillin-resistant S. aureus (MRSA). Virulence factors produced by S. aureus include hemolysins, leukocidins, proteases, enterotoxins, exfoliative toxins, and immune-modulatory factors (Oogai et al., 2011). In fact, extracellular toxins produced by S. aureus can cause toxic shock syndrome (G Abril et al., 2020).

The major antibiotic targets for staphylococci are the cell envelope, the ribosome, and nucleic acids (Foster, 2017). Commonly prescribed antibiotics used to treat S. aureus infections include cefazolin, nafcillin, oxacillin, vancomycin, daptomycin, and linezolid (Table 1) (Foster, 2017). These antibiotics are used because many strains of S. aureus are resistant to other traditional antibiotics through complex mechanisms (Tong et al., 2015). For example, S. aureus can acquire drug resistance through genetic mutations that modify the target DNA gyrase or reduce the number of outer membrane proteins (Tong et al., 2015). These mechanisms lead to a reduction in drug accumulation.

Aim

This study aims to develop a bioinformatics pipeline to discover unique and accessible ASO binding sites in the essential genes of three different bacteria (P. gingivalis, H. influenzae, and S. aureus) as a demonstration of its utility. By doing so, we can address the issues of antimicrobial resistance and off-target effects which plague traditional antibiotics. These binding sites will be prioritized based on their sequence-specificity, self-hybridization, melting temperature, and position relative to the open reading frame (ORF). Homologous regions were identified within the essential genes of the bacterium of interest to mitigate the ASOs’ off-target effect. This ensures that the ASO will not bind to the genes of humans and other organisms, circumventing the shortcomings of broad-spectrum antibiotics. Self-hybridization is defined as the process by which two identical, single-stranded ASOs form into a double-stranded molecule, depleting the availability for binding the target gene. Melting temperature and ASO stability are positively correlated. ASO sites that are near the start codon and in the upstream open reading frame (uORF) can effectively prevent translation by sterically blocking ribosomal subunit assembly or engaging mRNA degradation via RNaseH-dependent endonuclease mechanisms (Hegarty & Stewart, 2018). Currently, chemically modifying oligonucleotides is the approach to determine particular degradation mechanisms. We recommend interested readers to refer to literature such as Bonham et al. (1995). Therefore, ASOs that are specific, have low self-hybridization, high melting temperature, and bind near the 5′ end of the ORFs are preferable.

By applying these criteria to P. gingivalis, H. influenzae, and S. aureus, we found 1,117, 847, and 7,061 ASO targets respectively. The critical biological pathways identified include cell cycle, cell division, translation, and regulation of cell shape. The bacteria listed above are good candidates for RNA-based antibiotics because they are difficult to manage with traditional antibiotics due to the presence of biofilms and other antibiotic resistance mechanisms.

Materials and Methods

Bacterial essential genes

Essential genes defined as those required for the survival of an organism. Open reading frames (ORFs) of such genes were obtained from the Database of Essential Genes (DEG) (Zhang, Ou & Zhang, 2004). Relevant bacteria files regarding organisms, annotations, nucleic acids, and amino acids were downloaded from the DEG (tubic.org/deg/public/index.php/download). Among other organisms, the database included 66 bacteria with an average of about 403 essential genes per bacterium.

Gene (or ORF) sequences were obtained from the nucleic acids file (DEG10.nt.gz). Bacterial IDs were obtained from the bacteria file (deg_bacteria.csv). Additional information regarding each essential gene was obtained from the annotations file (deg_annotation_p.csv).

Homology

BLASTN (Basic Local Alignment Search Tool, version 2.13.0) (Camacho et al., 2009) searches were used to find homologous regions within the essential genes of the bacteria of interest, i.e., P. gingivalis, H. influenzae, and S. aureus. For system performance reasons, the homologous regions search was split into two. The first search compared the bacterium of interest with the human non-redundant nucleotide database by using the parameter “-taxids 9606”, where 9606 is the taxonomy ID of Homo sapiens. And the second search compared the bacterium of interest with all available non-human genomes by using the parameter “-negative_taxids 9606”. For the rest of the BLASTN parameters, default parameter values were used except word size was 7 and maximum Eval was 20 by which degenerate regions can be identified. Hits from different strains of a bacterium were eliminated. Examples of homologous hits can be found in Fig. 1. This is the most computationally intensive step of the entire pipeline. Each search (human or non-human) of a bacterium took approximately 4 days of runtime in a 20-core Intel® Xeon® Silver 4208 CPU @ 2.10 GHz with 32 GB on board memory. While the other steps mentioned below took less than 2 minutes to finish.

Figure 1 BLAST results.

BLASTN results. Query sequences (qseq) are highlighted in red. Search titles (stitle) are highlighted in blue. The top table contains columns one through seven. The bottom table contains columns eight through fourteen.

Secondary structure prediction

RNAfold (ViennaRNA-2.5.1) (Lorenz et al., 2011) and Mxfold2 (version 2.0.1.1) (Sato, Akiyama & Sakakibara, 2021) were used to predict the secondary structure for each essential gene. The secondary structure refers to the three-dimensional shape of the transcribed mRNA sequence. RNAfold predicted optimal secondary structure with nearest-neighbor free energy parameters while Mxfold2 predicted optimal secondary structure with nearest-neighbor free energy parameters alongside folding scores. Thermodynamic models were used to calculate the nearest-neighbor free energy parameter for both programs. Mxfold2’s folding score was calculated with a deep neural network (Sato, Akiyama & Sakakibara, 2021). It is noteworthy that Mxfold2’s folding score was unable to handle sequences with lengths greater than 1,000 base pairs. An example of a prediction is shown in Fig. 2. Two predicted secondary structures of ispF (DEG10220003) are exhibited in Figs. 2B and 2C. The gene ispF encodes 2-C-methyl-D-erythritol 2,4-cyclodiphosphate synthase.

Figure 2 Secondary structure predictions.

Secondary structure prediction result. (A) Line one is the header. Line two is the input sequence. Line three is the predicted secondary structure. “.” denotes an unpaired base; “(“ and “)” corresponds to the 5′ base and 3′ base, respectively. The number in parentheses is the thermal energy. (B) Visual representation of RNAfold prediction of DEG10220003. (C) Visual representation of MXfold2 results.

Masking

Regions suggested for homology may contribute to non-specific binding. Base pairs involved in the formation of stem structures indicate that they are inaccessible to ASOs. For the homology checking, nucleotide fragments of essential genes detected to be homologous to genes in other organisms were masked in lowercase letters. As shown in Figs. 3B and 3C, uppercase and lowercase fragments represent non-homologous and homologous regions, respectively. Figure 3A shows the original sequence.

Figure 3 Combined masking results.

Combined masking process and results. (A) Original essential gene sequence. (B) Essential gene sequence after masking human homology. (C) Essential gene sequence after masking non-human homology. (D) Essential gene sequence after masking regions involved with predicted secondary structures with MXfold2. (E) Essential gene sequence after masking regions involved with predicted secondary structures with RNAfold. (F) Final masked essential gene sequence. Headers were DEG IDs. Line one contained masked sequences. The final gene sequence involved overlaying masked homology and secondary structure sequences. The final masked sequence is highlighted.

Next, the semi-masked gene sequences, i.e., Figs. 3B and 3C, were further processed according to predicted secondary structures. Bases predicted by both RNAfold and Mxfold2 to be involved in the formation of stem structures, i.e., associated with parentheses, were masked. Figures. 3D and 3E show semi-masked gene sequences. Uppercase and lowercase fragments represent accessible and inaccessible regions, respectively. An example of the final masked sequence is shown in Fig. 3F. The above-mentioned process was repeated for human and non-human homology searches as well as for the two secondary structure prediction methods.

ASO generation

Antisense oligonucleotides (ASOs) were generated from the unmasked regions discussed above. The length of ASO is colloquially set to 19 or 20 base pairs (bps) as clinically tested ASO therapeutics are 20 nt or longer, e.g., Fomivirsen and Casimersen. If the sequence is too long, there is an increased chance of self-hybridization and increased non-specific binding. Long sequences have more regions that can partially hybridize with the target. Thus, increasing non-specific binding. On the other hand, sequences that are too short have low binding affinity and increased non-specific binding by chance. In this study, each ASO was 20 bps long (20 mer) and was a reverse complement of an unmasked region in the ORF (Fig. 4A). The output was saved as FASTA as illustrated in Fig. 4B.

Figure 4 ASO generation.

ASO generation from accessible (unmasked) regions. ASOs are reverse complements of unmasked sequences. (A) ASO generation of a sample sequence. (B) ASO sequences in FASTA format. Numbers at the end of the header indicate position relative to the start of the original gene sequence.

Self-hybridization and melting temperature

Each ASO was checked for the extent to which it hybridizes with itself, namely, self-hybridization (SH). We devised an SH score which is the maximum number of Watson-Crick pairings per sequence. An ASO sequence was aligned with itself in an antiparallel manner after shifting n positions, where n is 0, 1. 2.…, k/2, and k was the length of the ASO (Fig. 5). The algorithm stopped at k/2 shifts because matches beyond that point are rare. This scoring method assumes non-gapmer hybridization. MeltingTemp module of the Bio.SeqUtils package (https://biopython.org/docs/dev/api/Bio.SeqUtils.html) was used to predict the melting temperature and calculate the GC content of each ASO. Melting temperature (Tm) was calculated using the Tm_Wallace method. GC content was calculated using the Tm_GC method. For sequences between 14 and 20 bps in length, the Wallace rule is the standard mean to predict melting temperatures. Approximate melting temperatures (°C) are calculated based on the equation: Tm = 2(A + T) + 4(G + C), where A, T, G, and C refer to the number of each base pair in the sequence.

Figure 5 Self-hybridization score determination.

Visualization of self-hybridization score determination. k is the length of the ASO. The black sequence is the forward sequence. The red sequence is in the reverse sequence. The vertical lines represent Watson-Crick comparisons. The process is repeated until shift = k/2.

Putative potent ASOs were selected if they fulfilled all the following criteria: first quartile (Q1) SH score, fourth quartile (Q4) TM score, and within 100 bases of the start of the ORF (Figs. 6A, 6C, and 6E). All other ASOs were omitted. The details of the ASO sequences were integrated into a file per bacterium. A final file containing seven columns was generated. The column names include DEG ID, Position, ASO seq, SH score, Tmscore, GC score, and Gene name (Fig. 6G). The final results can be downloaded from GitHub: https://github.com/EricSHo/AOA/tree/master/data.

Figure 6 Self-hybridization scores vs. melting temperature.

Graphical representation of self-hybridization scores vs. melting temperature of a bacterium, ASO position, and the final CSV file. (A) It shows the distribution of self-hybridization scores and the melting temperatures of all ASO sequences from P. gingivalis. (B) It shows the distribution of ASO positions relative to the start of the ORF for P. gingivalis. (C) It shows the distribution of self-hybridization scores and the melting temperatures of all ASO sequences from H. influenzae. (D) It shows the distribution of ASO positions relative to the start of the ORF for H. influenzae. (E) It shows the distribution of self-hybridization scores and the melting temperatures of all ASO sequences from S. aureus. (F) It shows the distribution of ASO positions relative to the start of the ORF for S. aureus. (G) Final CSV file containing all selected ASOs, i.e., ASOs from the highlighted quadrant; position is 0-based and is the distance between the start of the ASO and the ORF. SH score is the self-hybridization score; TM score is the melting temperature; GC score is the GC composition of the ASO sequence; GO_BP contains GO IDs that are associated with biological processes; GO_MF contains GO IDs that are associated with molecular functions; cells with missing information are indicated with a “-”. The highlighted quadrant in (A), (C), and (E) represents the first quartile of self-hybridization scores and the fourth quartile of melting temperatures; the size of the points are proportional to the number of ASOs with those coordinates. The x-axis in (B), (D), and (F) is limited to be within 100 base pairs of the ORF.

GO annotation

Gene Ontology (GO) analysis was performed for all unique DEG IDs associated with the shortlisted ASOs mentioned above. DEG ID and GO ID associations were obtained from the file deg_annotation_p.csv downloaded from DEG. Only associations for the three bacteria of interest were extracted. As GO annotates genes into the following ontologies: Biological Process (BP). Molecular Function (MF), and Cellular Compartment (CC), we customed a Python script to split GO IDs into two ontologies (BP and MF) for each essential gene accordingly. These were stored in the columns GO_BP and GO_MF in the final output file (Fig. 6G).

Results

Bioinformatics pipeline

Figure 7 presents the workflow of the ASO target identification process. To develop potent antibiotics, the pipeline focuses on genes that are indispensable for the survival of bacteria. The Database of Essential Genes (DEG) was used as the source of essential bacterial gene sequences. That said, genes not listed in DEG are not necessarily unimportant. The essential gene sequences of a bacterium from DEG were put through a series of two filters: homology searches and predicted secondary mRNA structures. Homology searches were performed with BLASTN and served two roles. First, it ensured that similar sequences were not observed in humans. Second, it ensured that similar sequences were not observed in any other species. By doing so, we could identify unique sequences within the essential gene—increasing the specificity of ASO targets and mitigating off-target effects. Secondary mRNA structures were predicted with two independent methods: RNAfold and Mxfold2. Segments of the transcribed mRNA involved in the secondary structure were inaccessible to ASO sequences. Thus, base pairs involved in secondary structures would be poor targets. Two prediction methods were used to increase certainty in the results. Non-unique sequences and inaccessible base pairs were masked and stored in FASTA format.

Figure 7 Proposed bioinformatics pipeline.

Overall workflow of the proposed bioinformatics pipeline.

ASO candidates containing 20 nucleotides (20mer) were generated from the unmasked regions. A length of 20 nucleotides is the “sweet spot.” It is long enough for high sequence specificity while being short enough to minimize the formation of duplexes (Wu, Kriz & Sharp, 2014). Final ASO candidates were selected after filtering based on self-hybridization, melting temperature, and position relative to the ORF. Lastly, gene enrichment analysis was performed to identify the biological pathways impacted by ASO targets on specific genes. By doing so, we can identify the potential impacts of targeting such genes of a particular species through a literature review.

P. gingivalis results

P. gingivalis has 463 essential genes. Table 2 shows the key statistics regarding accessible regions and the number of ASO targets. In general, good candidates have low self-hybridization scores, high melting temperatures, and are close to the 5′ end of the ORF. We classified good candidates as being in the first quartile SH score, fourth quartile Tm score, and starting within 100 bases of the ORF (Figs. 6A and 6B). The first quartile SH score is 6 and the fourth quartile Tm score is 58 °C. Out of the 45,628 ASOs, 1,117 were considered putative targets.

Table 2 An overview of the putative targets from P. gingivalis, H. influenzae, and S. aureus identified through the bioinformatics pipeline.

Bacteria	Accessible region (%)	Number of unique essential genes	Number of ASOs	Number of qualified ASOs	Proportion of genes with ASOs (%)	
P. gingivalis	43.91	348	45,628	1,117	40.17	
H. influenzae	10.54	191	27,273	847	29.75	
S. aureus	89.11	346	175,606	7,061	98.58	

Selection of ASO candidates

We ranked ASOs based on the following characteristics: self-hybridization score, melting temperature, and position on the gene. ASOs with low SH scores have increased efficacy as they can bind better to the target sequence. The expected SH score per ASO was equal to P(SH) = ASO length × 1/4 = 5. This indicates that we expect 25% of the ASO length to bind to itself by chance. For P. gingivalis, the cut-off SH score in the first quarter was 6. Thus, all selected ASOs have SH scores less than 6. This means that we expect 33% of the ASO length to bind to itself by chance; leaving 67% to bind with the target sequence.

Melting temperature can be used to infer the stability of the binding between the ASO and the target gene sequence. The optimal primer pair melting temperatures are between 50 °C and 60 °C (“Primer Design for PCR”, Primer Design, 2023). Selected ASOs were in the fourth quartile. For P. gingivalis, H. influenzae, and S. aureus the cut-offs were 58 °C, 54 °C, and 54 °C, respectively.

ASO sites that are near the start codon, as well as those within the uORF, are more potent than those further downstream. Such sites can effectively prevent translation by sterically blocking ribosomal subunit assembly (Hegarty & Stewart, 2018) or trigger RNaseH-dependent degradation. Thus, as a general bioinformatics pipeline, we prioritized ASOs that began within 100 bases of the ORF after selecting sequences that fulfilled the SH score and Tm score requirements. Table 2 shows the number of qualified ASO candidates per bacterium.

Comparisons with two more bacteria

H. influenzae has 642 essential genes. After masking, we calculated that 10.54% of the gene sequence space was considered accessible. These accessible regions generated 27,273 ASOs from 191 unique genes (Table 2). Good candidates met the following criteria: being in the first quartile of the SH score, the fourth quartile of the Tm score, and within 100 bases of the ORF (Figs. 6C and 6D). The first quartile of the SH score was 6 and the fourth quartile of the Tm score was 54 °C. Out of the 27,273 ASOs, 847 were considered good candidates (Table 2).

S. aureus has 351 essential genes. After masking, we calculated that 89.11% of the gene sequence space was considered accessible. These accessible regions generated 175,606 ASOs from 346 unique genes (Table 2). Good candidates were determined as previously described (Figs. 6E and 6F). The first quartile of the SH score was 8 and the fourth quartile of the Tm score was 54 °C. Out of the 175,606 ASOs, 7,061 were considered good candidates (Table 2).

Gene enrichment analysis

Gene Ontology (GO) connects gene functions to three domains: biological processes, molecular functions, and cellular compartments. P. gingivalis has 95 unique GO ontologies for biological processes (GO BP) and 137 for molecular function (GO MF). Ontologies were priorities based on the count. The top 5 GO BP terms include: cell cycle (GO:0007049), cell division (GO:0051301), regulation of cell shape (GO:0008360), translation (GO:0006412), and terpenoid biosynthetic process (GO:0016114) (Fig. 8A). The top 5 GO MF terms include: ATP binding (GO:0005524), DNA binding (GO:0003677), zinc ion binding (GO:0008270), transferase activity (GO:0016740), and metal ion binding (GO:0046872) (Fig. 8B).

Figure 8 GO enrichment analysis of targeted bacterial genes in P. gingivalis.

Graphical representation of Gene Ontology results. BP = Biological Process; MF = Molecular Function. (A) P. gingivalis top 20 GO BP results. (B) P. gingivalis top 20 GO MF results.

H. influenzae has 51 unique GO ontologies for BP and 65 for MF. The top 5 GO BP terms include: transport (GO:0006810), establishment of competence for transformation (GO:0030420), regulation of cell shape (GO:0008360), iron ion homeostasis (GO:0055072), and cell wall organization (GO:0071555) (Fig. 9A). The top 5 GO MF terms include: ATP binding (GO:0005524), DNA binding (GO:0003677), metal ion binding (GO:0046872), transporter activity (GO:0005215), and transferase activity (GO:0016740) (Fig. 9B).

Figure 9 GO enrichment analysis of targeted bacterial genes in H. influenzae.

Graphical representation of Gene Ontology results. BP = Biological Process; MF = Molecular Function. (A) H. influenzae top 20 GO BP results. (B) H. influenzae top 20 GO MF results.

S. aureus has 156 unique GO ontologies for BP and 215 for MF. The top 5 GO BP terms include: translation (GO:0006412), cell wall organization (GO:0071555), regulation of cell shape (GO:0008360), peptidoglycan biosynthetic process (GO:0009252), and cell cycle (GO:0007049) (Fig. 10A). The top 5 GO MF terms include: ATP binding (GO:0005524), structural constituent of ribosome (GO:0003735), rRNA binding (GO:0019843), DNA binding (GO:0003677), and magnesium ion binding (GO:0000287) (Fig. 10B).

Figure 10 GO enrichment analysis of targeted bacterial genes in S. aureus.

Graphical representation of Gene Ontology results. BP = Biological Process; MF = Molecular Function. (A) S. aureus top 20 GO BP results (B) S. aureus top 20 GO MF results.

Importantly, the biological processes and functions mentioned above align with the mechanisms of actions of existing antibiotics. For example, tetracycline is a common antibiotic for treating gingivitis. It functions by interfering protein translation (GO:0006412: translation in Fig. 8A). Cefixime is used to treat H. influenzae and S. aureus infection by inhibiting cell wall synthesis (GO:0071555: cell wall organization in Figs. 9A and 10A). These examples have demonstrated that the bacterial genes identified by the proposed pipeline perform biological processes and functions that are also targeted by on the market treatments, reducing the effort in investigating new antibacterial mechanisms of actions.

Discussion

Interpretation of gene enrichment analysis

We observed many overlaps in critical biological processes between bacterial species. Regulation of cell shape was identified as a critical biological pathway in all three bacteria. Bacterial morphology is critical to the survival of a species because it helps them cope with and adapt according to external conditions (Young, 2007). Cell cycle and translation were identified as critical pathways in P. gingivalis and S. aureus. In contrast, cell wall organization was identified as a critical pathway in H. influenzae and S. aureus. Critical pathways unique to P. gingivalis include cell division, terpenoid biosynthetic process, and biosynthetic process. Critical pathways unique to H. influenzae include transport, the establishment of competence for transformation, and iron ion homeostasis. Lastly, the peptidoglycan biosynthetic process was critical and unique to S. aureus. To validate our results, we compared the pathways identified through our bioinformatics pipeline to common antibiotic targets for P. gingivalis, H. influenzae, and S. aureus.

Although periodontal disease is multifactorial, P. gingivalis stands out as a keystone species. In general, pathways targeted by antibiotics prescribed to patients include protein synthesis, DNA synthesis, and cell wall formation. This aligns with the critical pathways linked to ASO targets identified through our bioinformatics pipeline. An interesting pathway identified by our pipeline was the terpenoid biosynthetic process (GO:0016114). Terpenoids are a diverse class of organic compounds that derive from five-carbon isoprene units and are found in most living organisms (Reyes et al., 2018). They play important roles in a variety of cellular processes such as cell wall and membrane biosynthesis, electron transport, photosynthesis, and other processes (Boronat & Rodríguez-Concepción, 2015). Therefore, targeting this pathway may be an efficient way to combat P. gingivalis.

H. influenzae contributes to a variety of diseases and infections. General pathways targeted by antibiotics prescribed to combat H. influenzae infections include cell wall formation, protein synthesis, and DNA synthesis. These pathways align with many of those identified through our bioinformatics pipeline. However, with the increasing prevalence of ampicillin-resistant H. influenzae strains, alternate targets should be explored. An interesting pathway identified through our pipeline is the establishment of competence for transformation. Transformation increases genetic diversity among the species and may allow bacteria to incorporate exogenous DNA that contributes to antibiotic resistance. Thus, terminating this pathway may slow down the progression of novel antibiotic-resistant H. influenzae strains.

In the greater discussion on antibiotic-resistant bacteria, S. aureus is guaranteed to make an appearance. Drug resistance is acquired through processes including genetic mutations and the reduction of outer membrane proteins. Pathways targeted by antibiotics prescribed to address S. aureus infections include cell wall formation and protein synthesis. Critical pathways identified through our bioinformatics pipeline agree with the targets of current antibiotics. Many antibiotics inhibit cell wall synthesis via binding to penicillin-binding proteins (PBPs). If the number of proteins is reduced or if the 3D shape changes, those antibiotics are no longer effective. Thus, targeting cell wall formation at the mRNA level can bypass the limitations of current antibiotics.

Limitations of the proposed pipeline

This study has four main limitations. First, imputed essential genes must be on the DEG database. This implies that genes for bacteria of interest must be annotated and identified as essential through prior research. Second, predicted secondary mRNA structures were used to identify accessible regions within the transcribed mRNA. However, computational prediction of secondary mRNA structure is an ongoing challenge compared with the recent success of protein folding prediction such as AlphaFold (Jumper et al., 2021). The 20 amino acids have distinct biochemical characteristics enabling accurate secondary structure predictions. Moreover, there is a plethora of experimental data on protein structure. These can be fed into a machine learning algorithm to train the model to accurately predict protein structures. However, the biochemical characteristics of nucleic acids are less distinct. Furthermore, there are limited experimental data on mRNA structure (Lange et al., 2012). Thus, limited structures are available to train machine learning algorithms. To overcome this challenge, computational biologists are proposing the use of synthetic data to test deep learning models (Flamm et al., 2022). Lastly, biological validation was not performed. To overcome this limitation, this study utilized two secondary structure prediction methods to increase confidence in results.

Challenges of RNA-based therapeutics

There are many upsides to RNA-based therapeutics. Some of these include the ability to act on targets that were previously inaccessible to traditional small-molecule drugs, fast and cost-effective development, and the ability to rapidly alter sequences. Nevertheless, there are a few challenges associated with RNA-based therapeutics. The most notable include the rapid degradation of exogenous RNA by RNase, clashing charges between the RNA molecule and cytoplasmic membrane, and strong immune responses elicited by exogenous RNA (Damase et al., 2021). Furthermore, there are challenges specifically associated with RNA-based antibiotics, including penetrating target microorganisms, delivery, and biofilm penetration (Winkle et al., 2021).

One of the most significant challenges when delivering RNA-based antibiotics is penetrating the cell envelope followed by transferring the large and negatively charged RNA molecules across a hydrophobic and negatively charged cytoplasmic membrane. To combat this challenge, scientists modify the backbone and sugar molecules to give the ASOs more affinity to their target and stability (Damase et al., 2021). Furthermore, drug delivery is a challenge. Most of the current oligonucleotide therapeutics are delivered via local or hepatic modes (Roberts, Langer & Wood, 2020). The development of effective technologies for other delivery methods is a major goal for the field of oligonucleotide antibiotics. Another challenge facing antibiotics—as well as RNA-based drugs—is biofilms. Biofilms are clusters of microorganisms where cells are embedded in a self-produced matrix of extracellular polymeric substances (EPS) that are adherent to each other and a surface (biotic or abiotic) (Flemming et al., 2016). They are a protected mode of growth that allows cells to survive hostile environments (Flemming et al., 2016). Biofilms increase resistance to antibiotics and host immune cells by blocking access to the bacterial community (Sharma, Misba & Khan, 2019). Thus, to target bacteria within a biofilm, RNA-based antibiotics must be able to penetrate the barrier and be able to get to the lower layers of the biofilm before traveling across the cytoplasmic membrane and inhibiting translation.

Despite the challenges of ASO antibiotics, this platform is not infeasible. Development of efficient ASO delivery strategies for bacteria has received less attention than for eukaryotic cells (Hegarty & Stewart, 2018). Nevertheless, steps have been made to get closer to delivery across the bacterial cell membrane. One of the most promising advances is the inclusion of cell-penetrative peptides (CPPs) (Hegarty & Stewart, 2018). These peptides can facilitate translocation of molecules across the cell membrane with limited membrane damage (Hegarty & Stewart, 2018).

Conclusions

The proposed pipeline successfully identified putative ASO targets in three bacterial species (P. gingivalis, H. influenzae, and S. aureus). These targets can be traced back to critical biological pathways in each species. The database selected in this project represents important species in research and clinical settings. The developed pipeline has potential utility in identifying potent ASO targets in other pathogenic species. Moreover, great advancements in ASO drug delivery have been made in recent years. A promising delivery mechanism includes the use of nano-drug delivery vesicles (Gagliardi & Ashizawa, 2021). Such vesicles consist of a lipid bilayer with an aqueous compartment which allows for more stability, selective delivery, minimal toxicity, and greater biocompatibility (Gagliardi & Ashizawa, 2021).

From a broader perspective, this project has demonstrated the impact of bioinformatics on drug development. The rise of antibiotic-resistant bacteria is causing a wave of untreatable infections worldwide (Blair et al., 2015). Traditional small-molecule drug development lags behind the accelerated rate of antibiotic resistance. Thus, improving antibiotic development is critical to keeping up with the increased presence of resistant strains. RNA-based therapeutics provide a platform by which we can quickly and cost-effectively alter sequences to keep up with the emergence of resistant strains.

Additional Information and Declarations

Competing Interests

Author Contributions

Data Availability

The authors declare that they have no competing interests.

Erica S. Chen performed the experiments, analyzed the data, prepared figures and/or tables, authored or reviewed drafts of the article, and approved the final draft.

Eric S. Ho conceived and designed the experiments, performed the experiments, analyzed the data, authored or reviewed drafts of the article, and approved the final draft.

The following information was supplied regarding data availability:

Code and raw data are available at GitHub and Zenodo: https://github.com/EricSHo/AOA.

Erica S. Chen, & Eric S. Ho. (2023). In-silico Study of Antisense Oligonucleotide Antibiotics. Zenodo. https://doi.org/10.5281/zenodo.8352195.

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
