# Peer review of "In-silico study of antisense oligonucleotide antibiotics"

_PeerJ, doi:10.7717/peerj.16343_

## Round 0.1 · original submission · Major Revisions

Although this study underscores a significant advancement in the ongoing battle against antibiotic-resistant bacteria, a paramount concern in global health due to the dwindling development of new drugs, three reviewers still raised major and minor issues, which required appropriate responses and revisions from the authors.

1. [interfering protein translation] should be [interfering with protein translation].
2. [approaches to discover] should be [approaches to discovering].
3. [than eukaryotic cells] should be [than for eukaryotic cells].
4. For me, the Figure 1; BLASTN results are so small that it's hard to confirm their details.
5. The resolution and quality of Figures 2, 3, and 8 need to be improved. Please make adjustments.

Reviewer 1 ·

Basic reporting

The manuscript, authored by Erica and Eric, is commendably written, providing comprehensive background information and a detailed discussion about the current understanding of ASO treatment. Essentially, they establish a bioinformatics pipeline to identify potential ASO targets across three bacterial species. This pipeline integrates various aspects such as the specificity, stability, and potential candidates of the ASO sequences, which underscores its usefulness.

Experimental design

Presently, there is a significant void in literature concerning the development of a pipeline for generating ASOs that target potential genes for antibiotic purposes. This manuscript notably addresses this gap and fits squarely within the Aims and Scope of the PeerJ journal. Moreover, the methods are described with enough detail and clarity, thereby offering sufficient information for replication.

Validity of the findings

No comment of this part.

Additional comments

Certain literature points to the optical length of ASO in E. coli as being between 10-12 nt. With this in mind, it would be valuable for the authors to discuss potential variances in optical lengths for different organisms, and clarify which category their current pipeline falls into. Furthermore, given the incorporation of LNA in the formal design, it raises the question of whether Tm is a parameter that should be considered within this pipeline.

The authors' choice of translational inhibition over RNA decay in designing the ASO prompts further explanation. What is the reasoning behind this decision? Additionally, the choice to target within 100 bases of the Open Reading Frame (ORF) warrants an explanation. The ribosome footprint does not necessarily require this length, with existing literature suggesting that 50 bases may disrupt ribosome assembly.

The statement made between Lines 433-435 needs further clarification. Could the authors explain why an ASO target inhibitory sequence would have no effect on the translated protein? This seems counterintuitive, and a detailed explanation would greatly enhance the reader's understanding.

·

Basic reporting

The work presented by Chen and Ho offers an interesting approach to generating complementary RNA molecules to inhibit the protein translation of essential genes. The authors developed a pipeline with well-characterized tools. The manuscript is easy to read, and the writing is clear and concise. Overall, this is a very interesting manuscript with the only limitation that this is an emerging field, and several culprits still need to be overcome to develop this technology further.

Experimental design

I kindly request to clarify the following, do authors refer to putative genes, predicted genes, pseudogenes, or genes of unknown function? This reviewer thinks this is important since now there is an emerging idea that some genes with unknown function are relevant in specific environmental conditions that render them not only active but essential for survival. I recommend clarification in this regard. This reviewer links this comment with the criteria used by the authors to use genes from the DEG database, which is not contradictory but important to explain in the document's main body.
In line 213, please clarify if the nr-database or specific genomes were used. I understand that authors used all non-human databases, but I think specifying is important.
For this reviewer, using two RNA-fold predicting tools is a good decision. However, I think the authors should include the rationale for using either prediction in the methods section in the final pipeline.

Validity of the findings

I kindly request that the authors provide the approximate time it takes to design the ASOs to know the approximate complexity of the monumental work presented in the discussion section.
Regarding the GO report, I suggest using a better graphical representation of these results and highlighting the importance of no off-targets found using the reported pipeline.
Regarding the discussion, this reviewer suggests to the authors tone down the interpretation of the gene enrichment analysis, this is not an RNA-seq or microarray paper, and I recommend supporting the importance of cell processes identified for each bacteria with the corresponding reference.
The third limitation that the authors propose in their discussion is, in this reviewer's opinion, unnecessary. The proposed method will limit the translation of a protein regardless of its processing, I suggest removing this limitation.
For this reviewer, the strongest limitation of their method is the lack of experimental evidence that the designed ASOs are working in vivo. However, I think the technique is sound and is worth publishing for future experimental evidence supporting the authors' findings. I suggest authors discuss the differences between another method that has been recently published (doi: 10.1261/rna.079263.122.) with theirs and further support this work with perhaps this reference: doi: 10.1093/nar/gkac362. This is not work done by this reviewer.
Finally, this reviewer strongly believes that the authors have enough information in the literature to propose a delivery and stabilizing method for ASO. I suggest adding a paragraph in the discussion where they propose how the designed ASO can be used as therapeutic agent.

Additional comments

This reviewer has the following additional comments regarding the manuscript.
In the abstract, I recommend using in line 22 "These molecules hybridize with their…"
In the abstract line 24, please correct to Three bacterial species…
In line 29 of the abstract, I suggest "the position on the gene in an operon."
In the introduction, this reviewer encourages authors to discuss a bit of the new methods for protein folding prediction and their use in drug design. I know that protein crystallography is the method to follow, but predictive methods such as RoseTTAFold, AlphaFold, and DeepFold may shed light on the design of novel antibiotic molecules. On the same line of thought, at the end of the first section of the introduction (line 69), a small line of other tools, such as phages, is encouraged to include.
In the section "Antibiotics" lines 85 to 103, I kindly request adding a line regarding the modifications in the RNA used as ASO that renders their stability, protect them from degradation, and are needed to introduce the molecule in the target cell.
I suggest adding the year (either of access or of publications) of the resources that focus on a specific topic, such as "antibiotic resistance", "Staph infections".
Line 188, I think, is more suitable "at the 5' end" rather than "the beginning". This is also true in line 315.
In lines 189 to 190, this reviewer thinks that adding the number of target genes the identified ASO are distributed is desirable.
In line 210 please correct to P. gingivalis, H. influenza and S. aureus, the full name of the genus has been stated in previous lines. This has also to be corrected in line 291.
In line 255 please change to "as FASTA format and illustrated in"
In line 265 please correct to Tm between parentheses and in the other instances used.
Finally, this reviewer thinks that Figure 1 could be improved in resolution and quality. Figure 3 the F panel has a shadow that, in this reviewer's opinion thinks masks the reading of the sequence. Figure 4 also seems of low resolution, and finally, figure 8 seems too small to read; I suggest improving the distribution of the graphs so that information is more accessible/readable.

·

Basic reporting

The theoretical work by Chen and Ho, represents a well written and documented study aiming to develop a bioinformatics platform to identify new antimicrobial targets for antisense oligonucleotides for 3 important pathogenic bacterial species.
I have only minor concerns:
1) Besides S. aureus, why not other SKAPE bacteria (rather than P. gingivalis and H. influenzae) we used in the study? Since they are the major threat for human health according to WHO and others?
2) L. 59 and elsewhere, capitalize “Gram”.
3) L. 159 change “into two strains” to “into two kind of strains”
4) L. 190 “49” is this number right?
5) L. 210 and elsewhere, abbreviate the bacterial names since they were fully written previously.
6) L. 229 “ispF” all gene names must be in italics.
7) Figure 8, rather than present the data in that form it could be better to draw a figure containing a schematic representation of the bacteria and its metabolism and the targets found.
8) References: the bacterial and gene names should be in italics.

Experimental design

no experiments were done.

Validity of the findings

They are OK, although not experimentally validated yet.

Additional comments

None

Cite this review as

---

## Round 0.2 · accepted · Accept

Although no response was obtained from one reviewer, 2 other reviewers have approved this revised article for publication. I agree with them. This article was able to be considered for publication in this journal.

However, it should also be noted that one reviewer still raised some concerns about the quality of the figures. The figures are still showing a pixelated pattern, specifically Figures 3, 8, 9, and 10. I hope that the authors will do their best to improve the clarity of these figures.

Reviewer 1 ·

Basic reporting

In this manuscript, Erica and Eric provide comprehensive background information and a detailed discussion about the current understanding of ASO treatment. Essentially, they establish a bioinformatics pipeline to identify potential ASO targets across three bacterial species. This pipeline integrates various aspects such as the specificity, stability, and potential candidates of the ASO sequences, which underscores its usefulness.

Experimental design

Presently, there is a significant void in the literature concerning the development of a pipeline for generating ASOs that target potential genes for antibiotic purposes. This manuscript notably addresses this gap and fits squarely within the Aims and Scope of the PeerJ journal. Moreover, the methods are described with enough detail and clarity, thereby offering sufficient information for replication.

Validity of the findings

The pipeline to generate the ASO candidates is simple but logistic. The authors have addressed all my previous concerns, although some of them are open questions and the answers are not perfect. I'd suggest accepting the current version.

·

Basic reporting

This reviewer thanks the authors for their revised version and appreciates deeply that they have addressed all the concerns this reviewer had. Also, their positive insight is greatly valued. This reviewer thinks the manuscript is now ready for acceptance.

Experimental design

The methods section is now fully described, no further comments on this section.

Validity of the findings

With the modifications done by the authors, the paper shows a valid and robust method for designing novel anti-sense oligonucleotides that may help stop multi-resistant bacteria.

Additional comments

In the final version of the manuscript, authors need to remove a yellow highlight and an extra space in line 201 (Bonham et al., 1995).
This reviewer also feels that the figures (at least in the review material) is still showing a pixelated pattern, specifically Figure 3, 8, 9 and 10. This may be related to the software used or the file type. I think this has to be delt with the editorial team.